# Direct and Indirect Impact of COVID-19 for Patients with Immune-Mediated Inflammatory Diseases: A Retrospective Cohort Study

**DOI:** 10.3390/jcm10112388

**Published:** 2021-05-28

**Authors:** Valeria Belleudi, Alessandro C. Rosa, Francesca R. Poggi, Alessandro Armuzzi, Emanuele Nicastri, Delia Goletti, Andrea Picchianti Diamanti, Marina Davoli, Nera Agabiti, Antonio Addis

**Affiliations:** 1Department of Epidemiology, Lazio Regional Health Service, 00147 Rome, Italy; a.rosa@deplazio.it (A.C.R.); f.poggi@deplazio.it (F.R.P.); m.davoli@deplazio.it (M.D.); n.agabiti@deplazio.it (N.A.); a.addis@deplazio.it (A.A.); 2IBD Unit, Fondazione Policlinico Universitario A. Gemelli IRCCS, Università Cattolica, 00168 Roma, Italy; alessandro.armuzzi@policlinicogemelli.it; 3National Institute for Infectious Diseases, Lazzaro Spallanzani, IRCCS, 00149 Roma, Italy; emanuele.nicastri@inmi.it (E.N.); delia.goletti@inmi.it (D.G.); 4Department of Clinical and Molecular Medicine, Sant’Andrea University Hospital, Sapienza University of Rome, 00189 Rome, Italy; andrea.picchiantidiamanti@uniroma1.it

**Keywords:** COVID-19, immune-mediated inflammatory diseases, rheumatoid arthritis, inflammatory bowel diseases, psoriasis

## Abstract

Importance: Since the beginning of the Coronavirus Disease-19 (COVID-19) pandemic, Severe Acute Respiratory Syndrome-CoV-2 (SARS-CoV-2) infection has been a serious challenge for immune-compromised patients with immune-mediated inflammatory diseases (IMIDs). Objective: Our aim was to investigate the impact of COVID-19 in terms of risks of infection, hospitalization and mortality in a cohort of patients with rheumatoid arthritis (RA), psoriasis (PSO) or inflammatory bowel disease (IBD). Furthermore, we studied the impact of SARS-CoV-2 infection on the prescribed drug regimen in these patients. Methods: Through the record linkage between health information systems, a cohort of patients, ≥18 years old, assisted in the Lazio region and who had suffered from immune-mediated inflammatory diseases (RA, PSO, IBD) between 2007 and 2019, was identified. The risk of infection, hospitalization or mortality for COVID-19, was assessed by logistic regression models, and reported in an Odds Ratio (ORs; CI 95%), adjusting for sex, age and the Charlson Comorbidity Index. We also estimated these risks separately by IMID and in the subgroup of prevalent biologic drug users. We investigated deferral of biological treatments in the study population by comparing the prevalence of weekly use of biologicals (2019–2020) before and during the pandemic periods. Findings: Within the 65,230 patients with IMIDs, the cumulative incidence for COVID-19 was 303/10,000 ab. In this cohort of patients, we observed a significantly higher risk of SARS-CoV-2 infection than the general population: OR = 1.17 (95% CI 1.12–1.22). The risk was higher even considering separately each disease and in the subgroup of prevalent biologic drug users. This last subgroup of patients showed a higher risk of death related to COVID-19 (OR 1.89; 95% CI 1.04–3.33) than the general population. However, no differences in terms of risks of hospitalization or death related to COVID-19 were recorded in patients with the IMIDs. Comparing the 2019–2020 prevalence of weekly biological drug treatments in prevalent biologic drug users, we found a decrease (−19.6%) during the lockdown, probably due to pandemic restrictions. Conclusions and Relevance: Patients with IMIDs seem to have a higher risk of SARS-CoV2 infection. However, other than for patients with prevalent biologic drug treatment, no significant differences in terms of hospitalization and mortality were reported compared to the general populations; further investigation is warranted on account of unmeasured confounding. In addition, during the lockdown period, the COVID-19 emergency highlighted a lower use of biologic drugs; this phenomenon requires strict pharmacological monitoring as it could be a proxy of forthcoming long-term clinical progression.

## 1. Introduction

The coronavirus disease-19 (COVID-19) pandemic raises relevant concerns in patients with immune-mediated inflammatory diseases (IMIDs), in particular if they belong to a vulnerable population at an increased risk of COVID-19 prevalence and complications as a consequence of the intrinsic immune system dysregulation and the use of immunosuppressive therapy [1]. Although data are currently not straightforward due to the heterogeneity of the autoimmune diseases, study design and ongoing therapeutic regimen [2], IMID patients have been included in most countries among vulnerable groups, having priority for Severe Acute Respiratory Syndrome-CoV-2 (SARS-CoV-2) vaccine administration [3,4,5,6].

In this context, a main challenge is whether to modify/suspend the ongoing immunosuppressive treatment in IMID patients; clinicians indeed have to consider both the potential risk of an increased infectious exposure and that of disease relapse induced by therapy suspension. The management of IMID patients during the COVID-19 pandemic has become significantly more complex for other reasons such as the limited access to primary care or the contradictory messages on the protective/causative role of the immunosuppressor for SARS-CoV-2, which can fuel psychological distress and lead to a reduction in compliance and adherence to the immune-suppressor therapy [7]. For these reasons, we aim to study the risks of infections, hospitalizations, or mortality related to COVID-19 in a cohort of patients with rheumatoid arthritis (RA), psoriasis (PSO), or inflammatory bowel diseases (IBD). Furthermore, we want to study the prevalence of drug treatments with biologic therapies before and after the lockdown due to the pandemic period and in particular during the pandemic restrictions.

## 2. Methods

The Lazio region, with a land area of 17,242 km^2^, is one of the 20 administrative regions of Italy and is in the central peninsular area of the country. It has 5,864,321 inhabitants, most of them (4,353,738) resident in the metropolitan city of Rome.

This is an observational study based on regional administrative healthcare databases. All patients selected, aged ≥18 years and residing in the Lazio region, were covered by the regional healthcare system on 31 December 2019.

From the study population, patients affected by IMIDs (RA, PSO, IBD) were identified, retrieving specific diagnosis and co-payment exemption codes from claims data during 2007–2019. Furthermore, we identified biologic drug users treated for RA, PSO, and IBD during 2019 and 2020.

To evaluate the direct impact of COVID-19 on the study population a record linkage with the regional COVID-19 surveillance registry was performed (data updated to 15 December 2020). We compared the risks of infection, hospitalization, and mortality related to COVID-19 in patients affected by RA, PSO, or IBD with the same data in the general population. Data were estimated by logistic regression models, and reported in an Odds Ratio (ORs; IC 95%), adjusting for sex, age, and the Charlson Comorbidity Index (CCI) [8]. We also estimated these risks separately by disease and in the subgroup of prevalent biologic drug users.

Finally, the possible interruption or deferral of biological treatment use during the COVID-19 pandemic in patients with IMIDs previously treated with biological drugs was investigated, identifying all individuals with at least one administration during the time-window of interest. In particular, the prevalence of weekly biologic use during 2020 was compared with the same period in 2019 using a historic cohort selected following the same criteria adopted for the study cohort. Specifically, the percent variance (Δ%) between 2020 and 2019 were estimated for several periods: before and during lockdown and before and during the SARS-CoV-2 second wave.

## 3. Results

In the Lazio region (about six million of inhabitants), from the start of the COVID-19 pandemic until 15 December 2020, the incidence of SARS-CoV-2 infection in the general population has been estimated to be at 266/10,000 inhabitants. Table 1 summarizes the characteristics of patients with IMIDs, according to the diagnosis of RA, IBD, and PSO, compared to general populations. Here, we reported the data stratified for gender, age, CCI, and SARS-CoV-2 cumulative incidence per 10,000 inhabitants during 2020.

Table 1 shows the risks of infection, hospitalization, or mortality related to COVID-19 in IMID patients. In this cohort of patients, we observed a significantly higher risk of SARS-CoV-2 infection than the general population: OR = 1.17 (95% CI 1.12–1.22). This finding was confirmed when we analyzed data separately for the subgroup of IMIDs and for prevalent biologic drug users. No differences in terms of hospitalization or mortality related to COVID-19 were reported. However, in the subgroup of prevalent biologic drug users, a significantly higher risk of death related to COVID-19 roughly double that of the general population (OR 1.89; 95% CI 1.04–3.43) was reported (Figure 1).

Comparing the prevalence of weekly use of biologic drug treatments (Figure 2) during 2019–2020, a relevant reduction in all IMID patients during lockdown (Δ% = −19.6%) was reported, with a minimal difference between each category group, Δ% = −25.5%, −27.4%, and −18.0% respectively for RA, PSO, and IBD.

## 4. Discussion

This is the first study aiming to analyze both the direct and indirect impact of the COVID-19 pandemic in patients with IMIDs in Italy, integrating the data already available in other countries [9,10,11,12].

In our cohort, which included 65,230 patients with IMIDs, we showed that these patients have a higher prevalence of SARS-CoV-2 infection (303/10,000 inhabitants), than the general population. On the other hand, their clinical outcomes were not significantly different from that reported in the general population, with the exception of the subgroup of patients with prevalent biologic drug treatments in which a higher risk of mortality related to COVID-19 was detected. This could be theoretically related to an association between prevalent use of biological treatments and the severity of the disease. In a parallel analysis, we also showed a prevalent reduction of the weekly use of biologic drug treatments in patients with PSO, RA, and IBD during lockdown.

Regarding the direct impact of COVID-19 on patients with IMIDs, our results are similar to the already-published data, and with the EULAR and ACR recommendations for the management of the autoimmune rheumatic patients during the COVID-19 pandemic, which reported a slightly increased risk of infections and similar clinical outcomes to those showed in the general population [13,14]. Conversely, a systematic review and meta-analysis have recently reported that rheumatic autoimmune diseases represent a risk factor for poor outcomes in patients with COVID-19 [2], and a large Danish cohort study has found a lower rate, but a more severe disease course, of COVID-19 among patients with IMIDs compared to the background population [10].

In the current analysis, we also observed a general reduction in the weekly prevalence of biological drug treatments during 2020, which is particularly low, in all three IMIDs considered, in the lockdown period. These observations can have at least two explanations. Indeed, throughout the COVID-19 pandemic, physicians and patients have developed several concerns on the use of immune-modulating therapies in patients with IMIDs, mainly due to their ability to increase the risk of infectious disease. In addition, logistical difficulties in reaching the hospitals for continue therapy has been showen [7].

However, it is important to consider that among the disease-specific risk factors, the use of biologic and synthetic immunosuppressive agents (other than rituximab), were reported to have an indifferent or protective, rather than a negative, effect on COVID-19 outcomes in IMID patients [15]. On the other hand, having an active disease appears to be associated with poor COVID-19 prognosis [1,9,16]. Moreover, the interruption of biologics observed in indirect analysis may be responsible for the increased risk of death in prevalent biologic users.

Thus, as already stated by the above-reported international recommendations, IMID patients (without suspected or confirmed COVID-19) should be advised to continue their treatment unchanged [13,17].

The strengths of our study are in the large number and the population-based design; however, some limitations exist. First, our results come from a single Italian area; however, these epidemiological and clinical data from the Lazio region may be generalized for the whole country. Second, the higher risk for infection may be due to a higher attention to screen for SARS-CoV-2 in patients with autoimmune diseases that are known to be at a high risk of infectious disease [18]. In particular, a subgroup analysis showed that the proportion of patients with at least one test for COVID-19 in the IMIDs cohort was 25.2% versus 21.8% in the general population.

Finally, this is a preliminary analysis of data extracted from administrative records, and thus we have no information on some confounding factors such as BMI, disease severity, immunosuppressive and COVID-19-related therapies. Nevertheless, we reduced possible biases by estimating the risks and adjusted for age, sex, and CCI [8], a comorbidity index that includes several conditions, such as diabetes, cardiovascular, and pulmonary disease.

Clinical and epidemiologic data in the high-risk category of patients with chronic disease need to be strictly monitored considering the high risk of drug regimen discontinuation, SARS-CoV-2 infection, and of clinical deterioration. Because of the strong potential implications both for clinicians and for health care stakeholders, further research would be desirable from different geographical areas and health care organizations to evaluate the consistency, and over-time the stability, of our results.

## Figures and Tables

**Figure 1 jcm-10-02388-f001:**
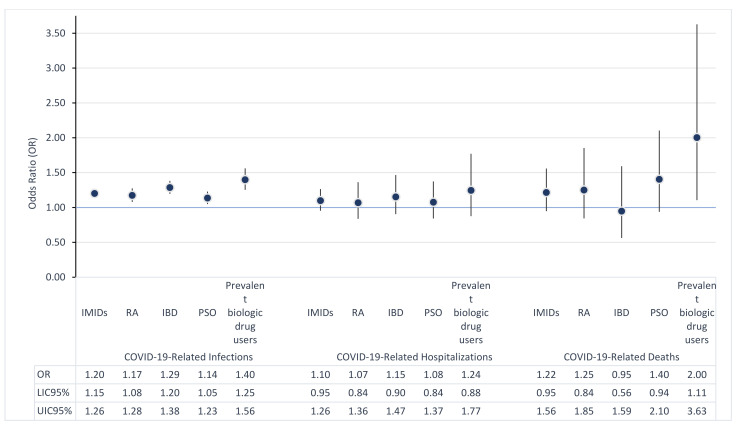
Risks of infection, hospitalization and mortality COVID-19 related in patients affected by IMIDs. Note. RA = Rheumatoid Arthritis; IBD = Inflammatory Bowel Disease; PSO = Psoriasis.

**Figure 2 jcm-10-02388-f002:**
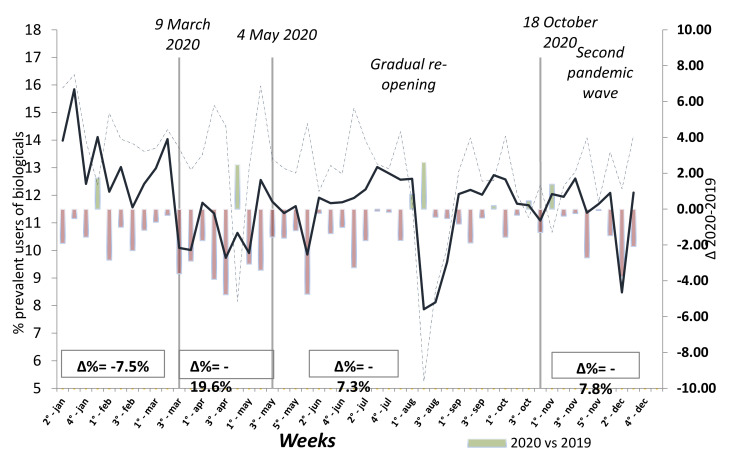
Prevalence of weekly use of biologic drug treatments in patients affected by IMDs previously treated with biologic drugs.

**Table 1 jcm-10-02388-t001:** Immune-mediated inflammatory disease (IMDs) cohort characteristics.

	IMIDs	RA	IBD	PSO	Prevalent Biologic Drug Users	General Population
	65,230	20,299	22,525	22,406	9176	4,702,567
Sex						
Male	42.4%	25.2%	52%	47.8%	42.9%	47.3%
Age (years)						
<50	29.9%	20.5%	41.3%	26.9%	35.3%	46.0%
50–60	21.3%	18.6%	22.1%	22.9%	25.4%	19.3%
60–70	21.5%	23.5%	17.4%	23.9%	23.2%	14.5%
70–80	17.2%	22.0%	12.4%	17.8%	13.2%	11.6%
>=80	10.1%	15.4%	6.9%	8.5%	3.0%	8.5%
Median [IQR]	59 (47–71)	64 (52–75)	53 (42–66)	60 (49–70)	56 (44–65)	51 (38–66)
Charlson comorbidity index						
0	80.7%	69.3%	87.7%	84.1%	83.1%	92.4%
1–2	14.4%	23.5%	9.1%	11.6%	14.8%	5.8%
3+	4.8%	7.3%	3.1%	4.3%	2.1%	1.8%
Covid-19 cumulative incidence during 2020 (per 10,000 inhabitants)	304	286	337	286	360	266

Rheumatoid Arthritis (RA); Psoriasis (PSO); Inflammatory Bowel Disease (IBD).

## Data Availability

The data that support the findings of this study are available from the Lazio Region but restrictions apply to the availability of these data, which were used under license for the current study, and so are not publicly available. Data are however available from the authors upon reasonable request and with permission of the Lazio Region.

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
