# Peer review of "Direct and Indirect Impact of COVID-19 for Patients with Immune-Mediated Inflammatory Diseases: A Retrospective Cohort Study"

_jcm, 2021, doi:10.3390/jcm10112388_

Round 1

Reviewer 1 Report

I read your manuscript with great interest.

The conclusions made are too strong with not enough scientific evidence in the manuscript. We are aware of several datasets including PsoProtect demonstrating clearly that an example of immune-mediated inflammatory diseases psoriasis per se do not have increased risk for infection with SARS-CoV-2 virus and complications due to COVID-19. But it is very important to mention that com-morbidities such as cardiovascular diseases , diabetes and pulmonary diseases as well as higher ager are associated with increased risk of complications from COVID-19.

In your manuscript these aspects as well as type of treatment (biologic, conventional systemic treatments etc.) are not studied at all. The dataset of 65230 patients is extremely valuable but the study must be done with detailed analysis to draw such a strong statement and conclusion.  

Unfortunately I do not find your manuscript suitable for publication in its current form.

Author Response

Reviewer #1

The conclusions made are too strong with not enough scientific evidence in the manuscript. We are aware of several datasets including PsoProtect demonstrating clearly that an example of immune-mediated inflammatory diseases psoriasis per se do not have increased risk for infection with SARS-CoV-2 virus and complications due to COVID-19. But it is very important to mention that com-morbidities such as cardiovascular diseases, diabetes and pulmonary diseases as well as higher ager are associated with increased risk of complications from COVID-19.

In your manuscript these aspects as well as type of treatment (biologic, conventional systemic treatments etc.) are not studied at all. The dataset of 65230 patients is extremely valuable but the study must be done with detailed analysis to draw such a strong statement and conclusion.  

Unfortunately I do not find your manuscript suitable for publication in its current form.

We thank the reviewer for the comment. We recognize that our study proposes just a preliminary analysis of the information extracted from administrative records. Indeed, we choose the “Brief report format” to rapidly provide these results to the clinicians and decision. However, several of the aspects raised by your concerns regarding co-morbidity, age and biological treatments are actually considered as adjustment in the present version. In fact, the risks estimated in our analysis were adjusted for sex, age and the Charlson Comorbidity Index. Even though. Although it has been done, residual confounding could have affected our results. Therefore, to better discuss it, we added a reference on Charlson Comorbidity Index and several sentences in discussion. Moreover, we added among the future developments, the aim to perform a more detailed analysis on the role of comorbidities.

Reviewer 2 Report

Overall well written and interesting- some points for consideration:

1) How does testing availability play into these results? Was everyone who was symptomatic getting tested at this time? I can imagine scenario where healthy folks with symptoms weren't getting tested if limited availability of tests, but patients with IMIDs were. This should be clarified, and what the testing protocol/availability in the region was during the study period. It's noted in the limitations but if any data available on this would include. 

2) How was biologic use assessed- as in, how do we know that patients deferred their therapy? Should be included in methods.

3) In Figure 1, are these biologic users including those who were deemed to have stopped their biologics during this time? This would impact interpretation, and may want to consider assessing only those who continued their biologics during this time since this is really the question that is trying to be assessed. Also significant literature suggesting no increased risk of death from biologic use- above point may be playing a role, or if not why do the authors think their results are different from others? 

4) Was BMI not able to be adjusted for? or baseline corticosteroid use which has shown to be associated with outcomes?

5) Limitations include not knowing the COVID-related therapies patients received during this time which should be included. 

Author Response

Reviewer #2

Overall well written and interesting- some points for consideration:

1) How does testing availability play into these results? Was everyone who was symptomatic getting tested at this time? I can imagine scenario where healthy folks with symptoms weren't getting tested if limited availability of tests, but patients with IMIDs were. This should be clarified, and what the testing protocol/availability in the region was during the study period. It's noted in the limitations but if any data available on this would include. 

We thank the reviewer for the comment. This is a good point and we added a new sentence in the discussion section on in depth analysis that compare the risk to be tested for Covid 19 between IMIDs patients and the general population.

2) How was biologic use assessed- as in, how do we know that patients deferred their therapy? Should be included in methods.

We thank the reviewer for the comment. We added details regarding this point in the methods section.

3) In Figure 1, are these biologic users including those who were deemed to have stopped their biologics during this time? This would impact interpretation, and may want to consider assessing only those who continued their biologics during this time since this is really the question that is trying to be assessed. Also, significant literature suggesting no increased risk of death from biologic use- above point may be playing a role, or if not why do the authors think their results are different from others? 

We thank the reviewer for the comment. We agree with the reviewer and therefore, we added details on the interpretation of our results in the discussion section.

4) Was BMI not able to be adjusted for? or baseline corticosteroid use which has shown to be associated with outcomes?

We thank the reviewer for the comment. We added details regarding residual confounding in the discussion section.

5) Limitations include not knowing the COVID-related therapies patients received during this time which should be included. 

We thank the reviewer for the comment. We added this point in discussion.

Round 2

Reviewer 1 Report

Dear Authors,

Thank you for your comments and re-submission.

Reviewer 2 Report

Comments addressed satisfactorily